# Minerals in the pre-settled coral *Stylophora pistillata* crystallize via protein and ion changes

Anat Akiva[1,6], Maayan Neder[2,3], Keren Kahil[1], Rotem Gavriel[4], Iddo Pinkas [5], Gil Goobes[4] & Tali Mass [2]

Aragonite skeletons in corals are key contributors to the storage of atmospheric $CO_2$ worldwide. Hence, understanding coral biomineralization/calcification processes is crucial for evaluating and predicting the effect of environmental factors on this process. While coral biomineralization studies have focused on adult corals, the exact stage at which corals initiate mineralization remains enigmatic. Here, we show that minerals are first precipitated as amorphous calcium carbonate and small aragonite crystallites, in the pre-settled larva, which then evolve into the more mature aragonitic fibers characteristic of the stony coral skeleton. The process is accompanied by modulation of proteins and ions within these minerals. These findings may indicate an underlying bimodal regulation tactic adopted by the animal, with important ramification to its resilience or vulnerability toward a changing environment.

[1] Department of Structural Biology, Weizmann Institute of Science, 76100 Rehovot, Israel. [2] Department of Marine Biology, The Leon H. Charney School of Marine Sciences, University of Haifa, 3498838 Mt. Carmel, Haifa, Israel. [3] The Interuniversity Institute of Marine Science, 88103 Eilat, Israel. [4] Department of Chemistry, Bar-Ilan University, 5290002 Ramat Gan, Israel. [5] Department of Chemical Research Support, Weizmann Institute of Science, 76100 Rehovot, Israel. [6] Present address: Laboratory of Materials and Interface Chemistry and Center for Multiscale Electron Microscopy, Department of Chemical Engineering and Chemistry and Institute for Complex Molecular Systems, Eindhoven University of Technology, 5600 MB Eindhoven, The Netherlands. Correspondence and requests for materials should be addressed to G.G. (email: gil.goobes@biu.ac.il) or to T.M. (email: tmass@univ.haifa.ac.il)

Recently, it has been shown that skeletal growth, in the adult *Stylophora pistillata* coral, involves transformation of amorphous calcium carbonate precipitate into an aragonite-made skeleton from centers of calcification that contain a skeletal organic matrix (SOM)[1,2]. A successful long-standing reef recovery requires better understanding of calcification capabilities of newborn corals under such harmful conditions[3]. The earliest stages of biomineralization may be the most vulnerable to changes in seawater chemistry[4,5]. Therefore, it is highly important to pinpoint the stage at which mineralization initiates and to identify regulatory mechanisms that may help protect the coral at its early stages of life.

Coral development involves two non-mineralized planktonic (free-swimming) stages; one on the first day after spawning, the planula swims actively and frequently changes its shape from spherical to pear-like, disk-like, and rod-like shape (Fig. 1a). The second stage is a metamorphosed globe-shaped mature larva which has six pairs of complete mesenteries (Halcampoides stage) (Fig. 1b). This larva subsequently changes into a benthic (settled) stage of a primary polyp (Fig. 1c), in which the formation of the aragonite exoskeleton is thought to be initiated[6–9]. Studies of skeletogenesis typically consider the primary polyp stage as the time point of the calcification onset[6]. To the best of our knowledge, no record of mineral formation during planktonic stages has previously been reported[6].

Here, we examine pre-settlement mineralization in corals, the minerals that are formed, and the role of distinct SOM proteins in this process. We used the Indo-Pacific stony coral *Stylophora pistillata*, a well-studied model coral, to investigate these questions. We applied a multidisciplinary approach that utilizes spectroscopic, imaging, and molecular biological techniques to study the early mineralization events prior to and immediately after coral settling just before extensive $CaCO_3$ mineralization occurs. We report the presence of amorphous calcium carbonate (ACC) and immature aragonite crystallites, in the planula stage of the coral. The earliest events of mineralization during the *S. pistillata* planula development are imaged using cryogenic scanning-electron microscopy (cryo-SEM) on pre-settled metamorphosed planulae and primary polyps a few days after settlement. The live planulae and primary polyps are high-pressure frozen and freeze-fractured prior to the cryo-SEM imaging, to ensure high-resolution measurements in a form as close as possible to the native state.

## Results

### Cryo-SEM and EDS analysis of pre-settled metamorphosed larva.
For the day-old pre-settled metamorphosed planulae, cryo-SEM shows extracellular deposits with sizes varying from one to tens of microns. These structures are identified by more intense backscattered electron (BSE) signals, which indicate that these deposits consist of elements of higher electron density, such as calcium (Fig. 2; Supplementary Figure 1). These deposits are randomly found within the organic material of the endoderm and lipidic region (see Fig. 2a). High-resolution images (Fig. 2d and Supplementary Figure 1) reveal that these deposits have a nanogranular structure. Additionally, the cryogenic energy-dispersive X-ray spectroscopy (cryo-EDS) (Fig. 2e–f) confirms the presence of elevated concentrations of calcium and carbon, and a depleted level of oxygen relative to the surrounding. Cryo-EDS also shows the presence of magnesium, which has implications for the mineral crystallization pathway[10]. Alongside such nanogranular deposits, mineral crystallites are observed with the typical acicular morphology of aragonite crystals (Fig. 2g–i).

Mineral deposits, both extracellular and intracellular, with a similar nanogranular structure were previously observed in different organisms and identified as ACC[11–13] which acts as either a transient precursor phase[11,14,15] or a sustained mineral storage phase[13,14,16], for subsequent formation of robust crystalline structures. Recently, amorphous particles were observed in the tissue of adult corals[2], suggesting that coral skeleton forms via amorphous precursor particles. However, in coral larvae, uncharacterized crystalline bodies ranging from 0.8 to 6 μm were previously observed in the gastrodermal epithelium in both pre-settled and primary polyp following fixation, decalcification, and osmication[17]. These structures were later suggested to be potassium chloride concentrating organelles[18]. No other evidence of mineral deposits in coral larvae was reported. This, however, is not surprising given that the organism mainly consists of organic tissue material and scarce amounts of newly formed mineral deposits that have not reached maturation. The presence of the minerals in this stage is further confirmed using spectroscopic measurements below.

### Cryo-SEM and EDS analysis of primary polyp.
For the primary polyp, the next stage of coral development, mineralization is observed to have spread extensively within the first week of settlement (Figs. 1c, 3; Supplementary Figure 2). The septa in Fig. 1c are also observed in the cryo-SEM image (Fig. 3a) along with round mineral deposits that are surrounded by the calicoblastic layer. A round deposit is depicted by an orange frame in Fig. 3a, of which SE and BSE images are shown in Fig. 3b and c, respectively. Under high magnification (Fig. 3d), these round deposits are seen to comprise acicular aragonite crystals which, in some cases, extend radially out of the nanogranular material in the center, similar to that observed in the pre-settled metamorphosed stage (Fig. 2i). These cryo-SEM images are reminiscent of a center of calcification (COC) with granular texture from which long needle-shape aragonite crystals extend (Fig. 3d), as shown for the adult stony coral before[1]. These findings support the hypothesis that initial granular mineral deposits serve as the building blocks for aragonite crystals that grow in a particle-attachment process[2], common to many biominerals[19,20]. The cryo-EDS analysis shows (Fig. 3e–f) that the mineral constituents calcium, oxygen, and carbon are prevalent. Interestingly, in contrast to the pre-settled metamorphosed stage, strontium and sulfur[21] are also present, but unlike in the planulae, no magnesium is detected. Higher-magnification cryo-SEM images of a septum (Fig. 3g–h), show that it is composed of developed aragonite blocks bordering nanogranular mineral. The transformation of a thin acicular structure into a bulky aragonite pillar structure of the septa walls is evident.

The presence of Mg ions in the early stage of amorphous mineral formation and of Sr ions in the later polyp stage in the region (seen in the cryo-EDS, Fig. 2e, f vs. Fig. 3e, f) where aragonite mineral accumulates may suggest a role of Mg in corals as a temporary inhibitory agent, stabilizing ACC during the mineral formation process and for Sr as a promoter of aragonite crystal maturation. Mg was previously shown to promote ACC and magnesium calcite (Mg-calcite) phases[22–24], while Sr was found in biogenic aragonite structures[25], especially during the early stage of $CaCO_3$ precipitation where it was found in high levels[26,27]. In the coral skeleton, Mg ions were shown to be distributed non-uniformly with particularly high concentration in the COC of more than 15 species[1,28,29], where ACC was found as well[30]. In line with Meibom et al.[28], the elevated concentration and homogeneous distribution of strontium, which is an abundant cation in the mature skeleton (~1 mol%), suggest that Mg and Sr are incorporated in the different mineral phases for very different purposes and by disparate pathways. While magnesium ions are transported via an extracellular pathway

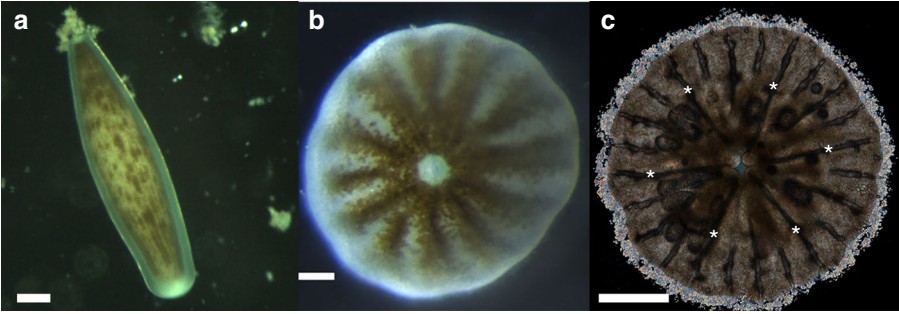

**Fig. 1** Light microscopy images of *Stylophora pistillata* larvae. **a** The pre-settled, elongated, planktonic larva (side view). **b** A globe-shape mature planula at the pre-settled metamorphosed stage. The brown features are attributed to endosymbiont algae. **c** The primary settled polyp (bottom view). Under polarized light, the birefringent crystals formed on the circumference appear as bright spots, while the mineralized septa[9], covered by organic layers of tissue, appear as the long dark streaks (the six primary septa are marked by asterisks), scale bars: **a**, **c** 500 µm; **b** 200 µm

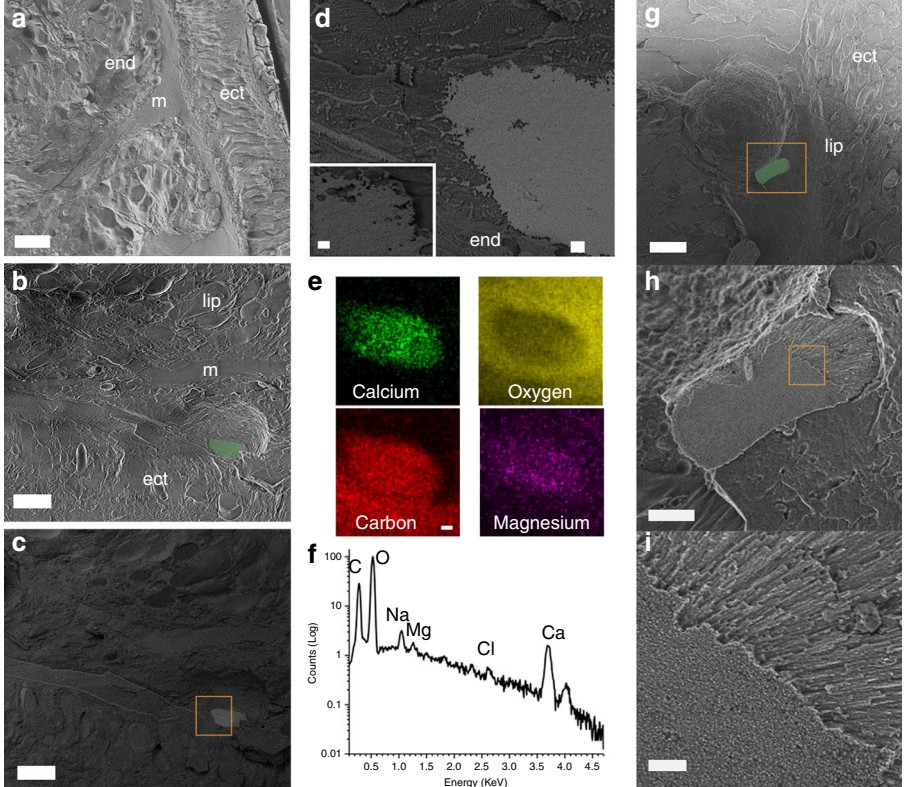

**Fig. 2** Cryo-SEM images of mineral deposits at metamorphosed planula. **a** Top view secondary electron (SE) image of a pre-settled metamorphosed planula. **b**–**d** Early stage of mineral deposition. **b** SE image of a mineral deposit randomly found within the organic material of the endoderm and lipidic region. The mineral deposit is ~10-µm long. It is digitally colored in green based on the backscattered electron (BSE) image in **c**. **c** BSE image of the same region as in **b**. **d** Higher magnification of the region depicted by an orange box in **c**, showing the mineral deposit morphology and the interface with the organic material of the tissue. The mineral is composed of nanogranular particles as can be seen in the insert in **d**. **e** Cryo-EDS maps of the mineral deposit in **b**, showing the distribution of calcium, oxygen, carbon, and magnesium in the mineral deposit and in the surrounding organic matter. **f** Cryo-EDS spectrum of the mineral in **b**. The spectrum is presented on a logarithmic scale. **g**–**i** A developing aragonite mineral embedded in the organic material of the tissue. **g** SEM image of a 10 µm aragonite crystal, colored in green based on the BSE image. **h** SE image of the region depicted in an orange box in **g**, showing the mineral. **i** Higher magnification of the region depicted in an orange box in **h**, showing that acicular aragonite crystals emerge from a nanogranular structure (end, endoderm; m, mesoglea; ect, ectoderm; lip, lipid gland). Scale bars: **a** 20 µm; **b**, **c**, **g** 10 µm; **h**, **e** 2 µm; **i** 400 nm; **d** insert, 200 nm

and/or vacuoles[31], strontium ions may be transported via the transcellular pathways, similar to the calcium pathway[32]. Whether Mg ions are transported out of the mineralization site as part of maturation is currently unknown.

**In vivo micro-Raman shows the existence of amorphous precursor**. For both developmental stages, in vivo micro-Raman

spectroscopy shows peak positions characteristic of aragonite (150, 203, 705, and 1085 cm$^{-1}$)[33] (Fig. 4a; see Supplementary Table 1 for complete peak assignment). The variations that appear in the low-wavenumber region (100–300 cm$^{-1}$) between a planula, a polyp, a mature coral branch, and geological aragonite, indicate a decreasing degree of atomic disorder in the mineral starting from the planula, to the primary polyp and finally the mature organism (Supplementary Figure 3 shows the low-

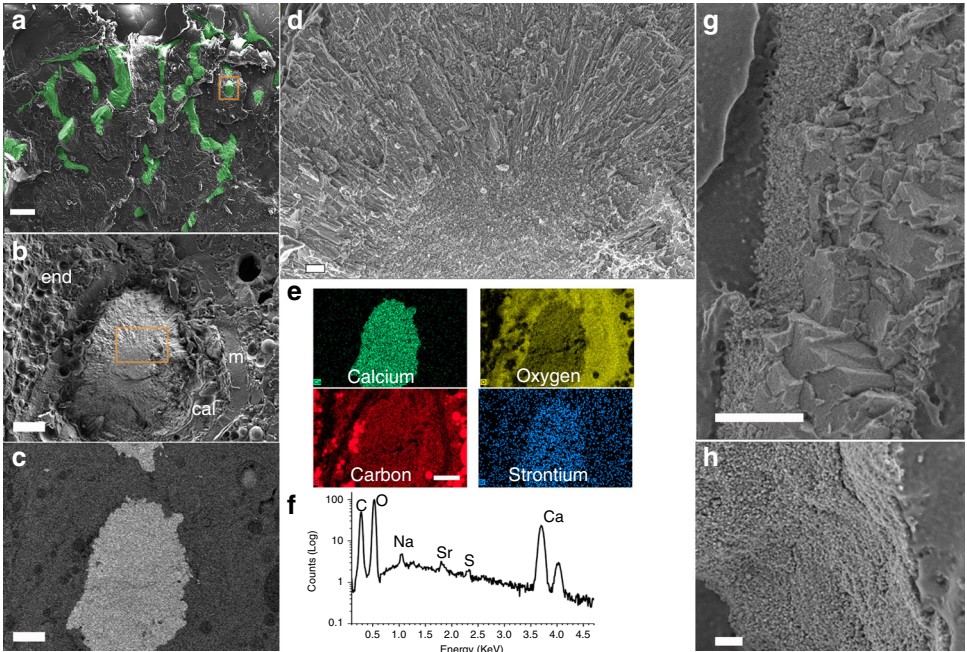

**Fig. 3** Cryo-SEM images of minerals at the primary polyp. **a** Low-magnification SE image of a primary polyp showing the part of the primary mineralized septa radially arrayed in hexacoral pattern (digitally colored in green). **b** High-resolution SE image of the mineral delimited by the orange box in **a** showing the mineral deposit surrounded by the calicoblastic layer (cal), mesoglea (m), and endoderm (end). **c** The corresponding BSE image of **b**. **d** High-resolution SE image of the mineral delimited by the orange box in **b**, showing the center of calcification with nanogranular structure from which long acicular aragonite rods extend. **e** Cryo-EDS maps of the mineral deposit in **b**, showing the distribution of calcium, oxygen, carbon, and strontium in the mineral deposit and in the organic material of the tissue. **f** Cryo-EDS spectrum of the mineral in **b**, presented on a logarithmic scale. **g** Exemplary SE image of a septum comprising two mineral morphologies, a region of nanogranular mineral, and a region of aragonite fibers. **h** Higher magnification of the nanogranular mineral of the septum. Scale bars: **a** 100 µm; **b**, **c**, **e** 10 µm; **d** 200 nm; **g**, **h** 1 µm

wavenumber region in more detail). In addition, an ACC phase was detected in several locations in the planula and in the primary polyp as well (Fig. 4a). This supports the cryo-SEM observation of two different mineral morphologies (Figs. 2 and 3).

**ssNMR distinguishes different nascent mineral phases**. The mineral phases and the protein–mineral interfaces were further characterized using solid-state NMR (ssNMR). Coral planula/polyps grown from [13]C-glucose-enriched seawater were measured in the different developmental stages. Direct excitation (DE) [13]C spectra of the pre-settled metamorphosed planulae (black) and primary polyps (red) exhibit a prominent $CaCO_3$ peak at ~171 ppm (Fig. 4b, c), evidencing that the bulk mineral phase is aragonite in both the pre- and post-settlement stages, consistent with the acicular aragonite crystallites observed in the cryo-SEM and Raman results. In the primary polyp, the carbonate line is more intense, indicating a larger content of aragonite, in accordance with the accelerated mineralization in this stage. The carbonate line also gets narrower in the polyp, indicating transformation from slightly disordered aragonite, in the planulae to more mature and better-ordered aragonite crystallites in the settled polyp. These differences in the mineral characteristics are inline with the presence of abundant aragonite fibers replacing the fewer and thinner needle-shaped aragonite crystallites (see carbonate peak analysis separately from the protein carbonyl peaks in Supplementary Figures 4 and 5).

In the proton-enhanced [13]C cross-polarization (CP) measurement, surface mineral phases are exclusively observed since bulk crystalline calcium carbonate has no hydrogen atoms in it. These spectra are therefore indicative of interfacial mineral deposits residing adjacent to hydrogen-bearing molecules. The [13]C CP spectra shown in Fig. 4d, e, clearly exhibit an ACC phase in the

pre-settled metamorphosed planulae (black) and primary polyps (red). An expansion of the carbonate region depicted by an orange box in Fig. 4d, is shown in Fig. 4e. The mineral peak indicated with a dashed line (Fig. 4e), is overlapping with the carbonyl carbon peaks from various proteins and requires mathematical analysis to separate out. In Supplementary Figure 6 and 7, we separate the carbonate peaks using deconvolution analysis. The carbonate peak at 169.3 ppm in the pre-settled metamorphosed planula is similar to the carbonate peak ascribed before to ACC in gastroliths[34]. The carbonate peak at 170.7 ppm in the primary polyps is similar to the carbonate peak ascribed before to either ACC at the aragonite surface in the skeleton of mature *S. pistillata*[1] and the nacre of *Haliotis laevigata*[35] or to disordered aragonite in the shell of *Perna canaliculus*[36]. The latter assignment better describes the state of the mineral, whereby the mineral peak shifts toward aragonite, suggesting initial arrangement of the ions into formation that is reminiscent of aragonite crystal lattice but with significant disorder due to immaturity of crystallites. The change from ACC to a better-ordered mineral phase leaning toward aragonite structure is in accordance with a metamorphic change in the coral[37]. The carbonate peaks are broader in both developmental stages than was reported before for biogenic ACC[34,38], as they represent nascent processes of mineralization not recorded before using NMR. Here, in the coral, the observed phases of ACC and disordered aragonite, appear separately in the two stages of development. The ACC in the pre-settled metamorphosed planulae and disordered aragonite in the primary polyp evidenced by the NMR are consistent with the nanogranules observed in the cryo-SEM (Figs. 2 and 3) and with the two mineral phases observed in micro-Raman (Fig. 4a). The nanogranular deposits undergo some atomic structuring during the transition to settled state and acquire some aragonitic character, as seen through the spectroscopic measurements.

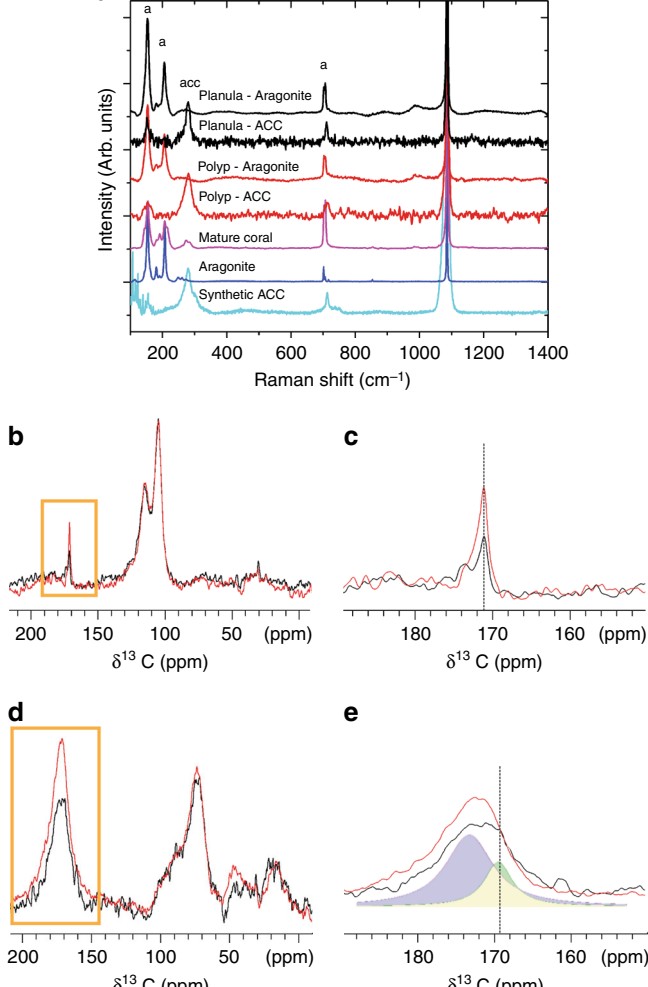

**Fig. 4** Spectroscopic mineral determination in the two stages. **a** Raman spectra of synthetic ACC (cyan), geological aragonite (blue), mature coral (pink), settled primary polyp (red), and pre-settled metamorphosed planula (black), showing the presence of the major aragonite bands and ACC. The main differences are in the low-wavenumber region (100–300 cm⁻¹) showing a clear signature of ACC in both early stages. **b**–**e** ¹³C ssNMR spectra of the entire pre-settled metamorphosed planulae (black) and primary polyps (red). **b** ¹³C direct excitation spectra of the two developmental stages. **c** The carbonate/carbonyl region of the spectra in **b**, showing the marked increase in the content of the bulk mineral after settlement. **d** ¹³C cross-polarization spectra of planulae and polyps enhancing the carbons with vicinal protons. **e** Magnification of the carbonate/carbonyl region of the spectra in **d**, showing changes of the disordered CaCO₃ line upon settling due to maturation of the carbonate ions. Dashed lines mark the location of the carbonate peaks in **c** and **e**. Peak analysis can be found in supporting information (Supplementary Figures 6 and 7). In the Raman spectra, typical peaks of aragonite (a) and ACC (acc) are indicated

Whether Mg ions are present in the ACC phase or not, as eluded to from the cryo-SEM, cannot be determined by the current NMR measurements. However, it was previously indicated that the biogenic ACC characterized by a carbonate resonance at ca. 169.3 ppm contains inorganic ions, including Mg[34].

The changes taking place between the two developmental stages lead to bulk mineral organization into a structure which constitutes mature aragonite blocks more closely. The

nanogranular deposits of completely disordered mineral are also more developed in the primary polyp and show characteristics of a precursor phase of crystalline aragonite. Maturation processes taking place both in the bulk and in the surface/interfacial layers indicate the mineral transformation undertaken by the organism as part of its metamorphosis.

**Skeleton organic matrix genes expression pattern**. The changes observed in the mineral deposits crystallinity and their relative quantity between the two stages suggest underlying activity of specific genes that are associated with mineralization[39,40]. Therefore, we analyzed the relative expression pattern of distinct "toolkit" genes, including four coral acid-rich proteins (CARPs 1–4) and carbonic anhydrase (STPCA2)[39–41]. This analysis revealed that CARP2, which is rich in glutamic acid, was up-regulated at the two planktonic stages. In contrast, CARPs 1, 3, and 4, which are rich in aspartic acid, and STPCA2, were mostly expressed after settlement (Fig. 5a).

Similar expression patterns were previously observed in *Pocillopora damicornis*[42], alluding to the possibility that this is a general phenomenon of coral mineralization and that glutamic-rich proteins may delay or retard crystal growth in corals before settlement. Aizenberg et al.[16] show that GlX-Ser and Gly-rich proteins induce the formation of stable ACC, both in the calcareous sponge *Clathrina* and in the ascidian *Pyura pachy-dermatina*[15]. In contrast, AsX-rich proteins were associated with crystalline CaCO₃[15].

**Atomic colocalization of minerals and proteins by ssNMR**. Next, we profiled the most abundant residues in each of the CARPs and STPCA2 proteins. The four most prevalent amino acids are summarized in Supplementary Table 2. On the basis of their expected prominence in the ¹³C spectra, we can envisage the dominant carbon peaks in ¹³C NMR spectra of the organism. As will be shown below, it allows us to link between patterns of actual protein upregulation and downregulation and those derived from gene analysis in the quantitative PCR (qPCR) results in different developmental stages of the animal.

Evidence of changes in proteins, carbohydrates, or other functional molecules prevalent in the two developmental stages can be drawn from their representative peaks in the 2D ¹H–¹³C HETCOR spectra superimposed in Fig. 5b. The spectra map the magnetization transfers between protons and adjacent carbons and indicate the most abundant carbons in each of the stages. By identifying peaks that are present in one spectrum and are absent in the other and vice versa, we can distinguish between unique molecules found in one stage but not in the other. Unique peaks are observed in the primary polyp at 34, 35.5, and 50 ppm (red) and in the pre-settled metamorphosed planulae at 22, 43, and 68 ppm (black) in Fig. 5b. The unique peaks in the polyp spectrum (red) represent the most abundant residues in CARP1, CARP3, CARP4, and STPCA2, while the unique peaks in the pre-settled metamorphosed planula spectrum (black) represent the abundant residues in CARP2 (see the Supplementary Information for additional information). In addition, the Cδ carbons in aspartate residues (177–179 ppm) are observed only in the primary polyp, as seen in the carbonyl/carbonate region of these 2D ¹H–¹³C HETCOR spectra (see Supplementary Figure 8), suggesting an increase in Asp-rich proteins in the polyp. Overall, the spectral differences are consistent with a decrease of CARP2 level and an increase in the levels of other CARP proteins in the polyp, assuming that changes to the levels of these biomolecules would be the most prominent spectral changes. This is in accordance with upregulation/downregulation of their associated genes, as inferred from the qPCR (Fig. 5a).

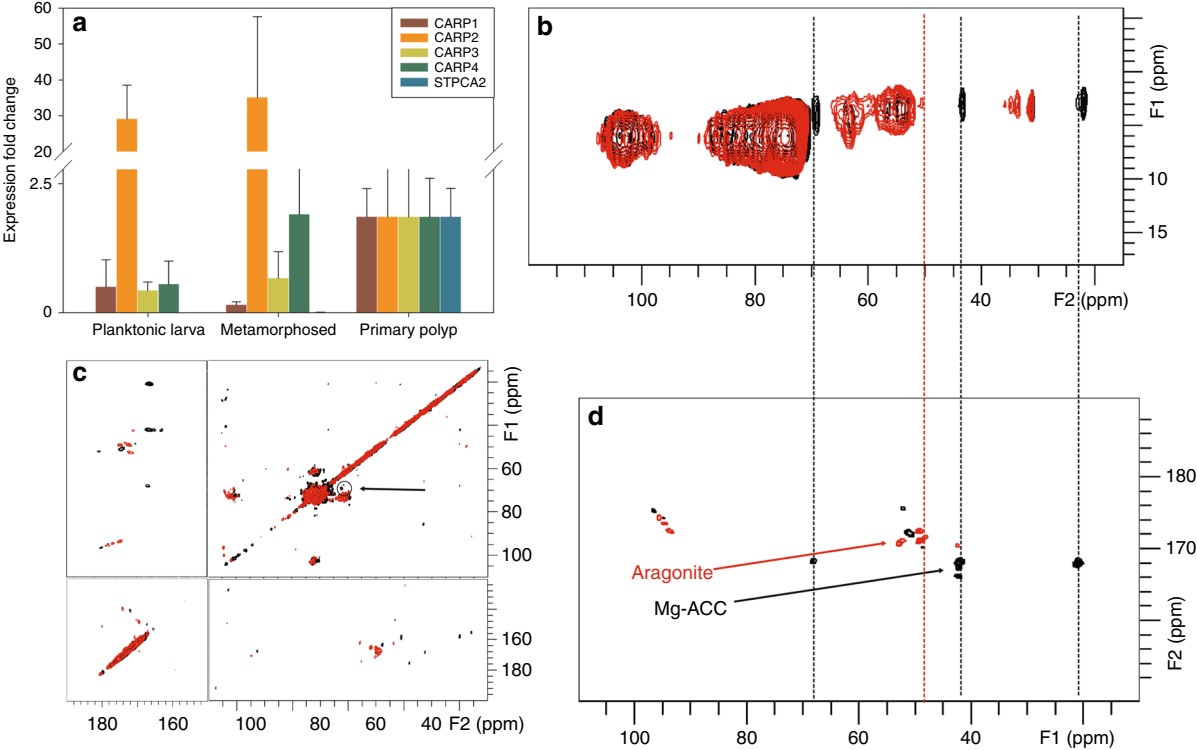

**Fig. 5** Identification of functional protein in the two stages. **a** qPCR analysis of the relative expression levels of distinct skeletal organic matrix genes determined by the $\Delta\Delta C_T$ method (see Methods section). The expression fold change is relative to the expression in the settled primary polyp stage. All samples were collected in triplicate and the results are presented as the average fold change ± SE, $n = 3$. **b** 2D $^1$H–$^{13}$C HETCOR spectra of pre-settled metamorphosed planula (black) and primary polyp (red). Spectra were recorded using a contact time of 2 ms, recycle delay of 1 s, and 4000 scans. **c** 2D $^{13}$C DARR NMR spectra of the entire pre-settled metamorphosed planulae (black) and entire primary polyps (red). The 2D DARR spectra show a large diagonal ridge and several off-diagonal peaks, appearing as dots in the 2D maps. These peaks indicate magnetization transfers between two different carbons. The peak at (F2, F1):(62, 68) encircled and marked by the arrow, indicates the magnetization transfer between threonine Cβ–Cα carbons. It is unique and characteristic of a protein with Thr prevalent in its sequence. **d** 2D $^{13}$C DARR (carbonate region)–blow-up of the carbonate region (upper left fraction in Fig. 5c rotated 90° counterclockwise) showing the strong correlations of the organic carbons with the disordered mineral phase at 169.5 ppm in the pre-settled planula metamorphosed and the weak correlations with the disordered aragonite at 171 ppm in the primary polyp. Dashed lines correlate common carbons peaks in the HETCOR (**b**) and in the DARR spectra (**d**)

Next, we implement 2D $^{13}$C DARR NMR experiments (Fig. 5c, d) to probe the proximity between any pair of carbon atoms (from proteins, carbohydrates, and carbonates) in the sample, that are within atomic distance from each other (within a distance lesser or equal to ~6 Å from each other). An expansion of Fig. 5c, which focuses on carbons that are proximate to the mineral carbonates, is shown in Fig. 5d. The peaks from pre-settled metamorphosed planula (black) indicate organics proximate to the Mg-ACC carbonates. The peaks from the primary polyp (red) indicate the organics near the aragonite carbonates. We show, using the dashed lines as guidelines that the exclusive carbon lines in the 2D HETCOR spectra in Fig. 5b are directly correlated to the respective mineral phases found in the two developmental stages. That is, the disordered carbonate peak at 168.3 ppm in the pre-settled metamorphosed planula is correlated with the CARP2 carbons at 22, 43, and 68 ppm and in the primary polyp, and the aragonite carbonate peaks at 170.4 and 170.7 ppm are correlated with carbons at 49–50 ppm of aspartate-rich proteins, i.e., CARPs 1, 3, and 4.

## Discussion

We show that during the coral *S. pistillata* development, mineral is formed while the planula is still in the pre-settled metamorphosed stage, which is earlier than what is currently thought. Having disordered mineral (ACC) distributed in the animal body,

may explain the rapid calcification of coral skeleton at its most vulnerable phase, the settlement. Moreover, by using a combination of molecular biological tools, advanced electron microcopy, micro-Raman spectroscopy, and NMR spectroscopy, we show the involvement of different proteins and ions in the two different developmental stages of the planula, exposing with detail the intimate interaction of the mineralization protein CARP2 with the ACC and the intimate binding of another set of active proteins (CARP1, CARP3, and CARP4) to the aragonitic phase that develops upon settlement. We further note that the role of other biomolecules in calcification is yet to be determined. As in many other calcifying organisms[11,43–45], coral skeleton organic matrix also contains polysaccharides and lipids[46–48]. Evidence for the intimate interaction of other biomolecules with the mineral carbonates in the settled polyp is seen, for example, in the solid-state NMR cross-peaks (Fig. 5d) between aragonite carbonates and aliphatic carbons at 92–96 ppm. However, the exact assignment of these carbons to lipids or polysaccharides is more complex and is outside the scope of the present study.

We suggest that in the pre-settled metamorphosed planulae, upregulated CARP2 is in atomic contact with Mg-ACC in the nanogranules. The nanogranules which are within the organic material of the endoderm, may have a role in the subsequent mineralization of the skeletal elements. In the settled polyp, the other CARPs and STPCA that are upregulated are in atomic contact with a more mature mineral phase, i.e., aragonite in

round deposits that resemble COC in mature corals. These proteins may be involved in promoting aragonite maturation and may accelerate mineralization. Selective localization of different CARP molecules within disparate mineral phases in the organism, provides a direct evidence of the differential functionality of these proteins in regulating the minerals formed and transformed, asserting some biological control over nascent mineralization processes in the animal.

These proteins are implicated with control of skeletogenesis by preorganization of the inorganic ions into an emerging mineral phase and into aragonite fibers that are similar to the ones making up its skeleton. Furthermore, certain CARPs are putatively used to stabilize an ACC mineral state during the motile stage, rather than to inhibit mineralization entirely, so as to allow a subsequent prompt transformation into aragonitic crystallites in the settled stage. The ability of coral recruits to rapidly calcify and transform the mineral may be aided by the temporal modulation of the CARP proteins and may thus have an important role in reef resilience and is a key factor in driving the recovery of coral reefs after disturbances such as ocean acidification. Therefore, an early onset of mineralization may enhance survival propensities for certain but not for other members of the scleractinia family.

## Methods

**Planulae collection and settling**. Planulae traps were created using a 160-μm plankton net, the top of which is attached to a plastic container. The planulae were collected from adult *S. pistillata* colonies under a special permit from the Israeli Natural Parks Authority in front of the Interuniversity Institute of Marine Biology in Eilat (IUI). The nets were placed on 14 adult corals for several nights during February–June 2016, following peak release[49]. Actively swimming metamorphosed larvae were analyzed within 24 h after collection. Individual planulae were maintained under ambient conditions (~25 °C and ~pH 8.2) in a flow-through outdoor aquaria exposed to natural lighting which received fresh seawater filtered to 60 μm. The metamorphosis and settlement processes proceeded normally on preconditioning treated microscope slides with crustose coralline algae (CCA). Primary polyps were collected 3 days after settlement.

**Synthesis of amorphous calcium carbonate for Raman measurements**. ACC was prepared by mixing 1 mL of 100 mM calcium chloride solution with 1 mL of 100 mM sodium carbonate (reagents by Sigma). Immediately after mixing, the solution was filtered with membrane filter, washed with absolute ethanol, and dried under an IR lamp. The dried precipitate was kept in a desiccator until it was measured[20].

**Cryo-scanning-electron microscopy (SEM) imaging**. Pre- and post-settled planulae were high-pressure frozen while they were still alive: individual animals were sandwiched between two metal disks (3 mm in diameter, with 0.1-mm and 0.05-mm cavities) and cryoimmobilized in a high-pressure freezing device (HPM10; Bal-Tec). The high pressure, applied during high-pressure freezing at 2000 atm for a few milliseconds, prevents water crystallization during cooling. The frozen samples were mounted on a holder under liquid nitrogen and transferred to a freeze-fracture instrument (BAF 60; Bal-Tec) by using a vacuum cryotransfer device (VCT 100; Bal-Tec), in which they were freeze-fractured at a temperature of −120 °C in a vacuum greater than $5 \times 10^{-7}$ mbar. Pre- and post-settled corals (10 and 5, respectively) were then transferred to a Zeiss Ultra 55 SEM where they were observed using a secondary electron in-lens detector and a backscattered electron in-lens detector (operating at 1.5 kV at a working distance of 2.2 mm). The samples were kept in the frozen–hydrated state at all times, using a cryo-stage at a temperature of −120 °C. To remove a thin layer of amorphous ice by sublimation and to expose the organic and inorganic content, the samples were etched by increasing the cryo-stage temperature inside the microscope to −105 °C for 10 min prior to imaging.

**Cryo SEM/energy-dispersive X-ray spectroscopy (EDS) analysis**. Elemental analysis and mapping were carried out at the same location by an EDS detector placed on the same optical axes as the SEM microscope. The loci of interest underwent cryo-EDS analysis using the microscope at a working distance of 6.5 mm, the spot size of 300 nm, dwell time of 8 μs, and an acceleration voltage of 8 kV, using a Bruker Quantax microanalysis system with an AXSXFlash® detector. Element distribution maps were obtained using the Quantax software. The identification of the Ca, Mg, Cl, C, O, and Na is based on the K edge, while the Sr identification is based on the L edge. The EDS map brightness and contrast levels were adjusted using Adobe Photoshop. The primary polyp was covered by a 6-nm carbon layer before the EDS measurement.

**Quantitative PCR (qPCR)**. Triplicate samples for qPCR were collected at the three developmental stages (i.e., two pre-settled metamorphosed and one post settlement; Fig. 1). The samples were snap-frozen in liquid nitrogen and stored in 0.55 ml of TRI reagent (Life Technologies) at −80°C until the RNA extraction. The RNA was extracted using the TRI reagent (Life Technologies) following the manufacturer's protocol with some modification at the homogenization step, as described by Mass et al.[42]. We performed on-column DNase digestion using the PureLink RNA mini kit (Anbion®). To be able to compare the expression of the samples, the same RNA concentrations were used in the reverse transcription (RT) reaction, using SuperScript™, following the manufacturer's protocols. For all samples, another reaction without the RT was performed, to ensure that only DNA-free cDNA was used in the qPCR. The samples were analyzed by PCR with the primers set, as illustrated in Supplementary Table 3. Whenever it was possible, the primers were located on different exons. Otherwise, minus reverse transcriptase reactions were used to ensure that the genomic DNA signal can be neglected (more than 1000-fold). The transcript levels were determined by using the StepOnePlus System qPCR thermal cycle. Primers specific to CARPs 1–4, STPCA2, and the housekeeping gene actin were designed using the IDT's PrimerQuest© online tool (http://eu.idtdna.com/PrimerQuest/Home/About). The primer sequences were based on validated sequences of cDNA originating from local *S. pistillata* RNA[40,41]. All reactions contained the qPCR SYBR®Green master mix and the specific primers set. The thermal profiles were comprised of the hold stage (20 s in 95 °C), the PCR stage (3 s in 95 °C and 30 s in 60 °C for 40 cycles) and the melt-curve stage (15 s in 95 °C, 1 min in 60 °C, and 15 s in 95 °C). The expression level of the CARP genes was determined by the $\Delta\Delta C_T$ method[50]. The $C_{TS}$ of each gene were standardized to those of the housekeeping gene (actin) and then to those of the settled stage before the natural log transformation.

**Raman spectroscopy**. Sample preparation: For pre-settled metamorphosed planula imaging, the planulae were briefly anesthetized with 5% MgCl, mounted on a microscope slide in seawater, and examined directly (five individual samples). Post-settled polyps (20 individual samples) were imaged in seawater on the microscope slide they settled on. At the end of the experiment, all planulae/polyps were still alive and apparently in good condition. Raman measurements were conducted on a LabRAM HR Evolution instrument (Horiba, France) configured with four laser lines allowing for Raman spectra from $50 \text{ cm}^{-1}$ and onward. The instrument is equipped with an 800-mm spectrograph which allows for pixel spacing of $1.3 \text{ cm}^{-1}$ when working with a 600 grooves/mm grating at 632.8-nm excitation.

The sample is exposed to light by various objectives (LUMPLFLN N.A.–−1.0, ×60 water immersion, Olympus, Japan was used in most of the experiments). The LabRAM instrument is equipped with two detectors: a $1024 \times 256$-pixel open-electrode front illuminated with cooled CCD camera. The system is set around an open confocal microscope (BX-FM Olympus, Japan) with a spatial resolution better than 1 μm using a ×100 objective. Most of the work on this project was performed using the 632.8-nm HeNe laser, the 600 grooves/mm grating, and the ×60 objective.

Exposure was set according to the signal intensity, and normally was set below 60 s. Exposures between 15 s and 10 min were used.

**NMR spectroscopy**. The holobiont (planula and zooxanthella) were fed with 3 g L$^{-1}$ (D-$^{13}$C$_6$-glucose, 99% $^{13}$C) for 7 days, generating $^{13}$C-enriched carbon-containing species (molecules, ions) that formed through the metabolic breakdown of the sugar. To avoid settlement, the planulae were incubated in 0.2-μm filtered seawater, while, to induce settlement, crustose coralline algae were added to the growth medium. The planulae were collected from the solution by spin-down and the primary polyps were scraped off the Petri dish using tweezers. Eight pre-settled metamorphosed planula and five post-settled polyps were harvested by dipping in pure ethanol to reduce salt concentration necessary to run NMR experiments with high-power decoupling. The planulae and polyps were then promptly washed in D$_2$O and were immediately freeze-dried. They were packed inside a Kel-F insert (Bruker Kit B4493) and inserted into a MAS NMR rotor (Supplementary Figure 9). Approximately 1.5($\pm 0.1$) mg of pre-settled metamorphosed and 2.0($\pm 0.1$) mg of settled forms of corals were packed into rotors.

Additional measurement of the pre-settled metamorphosed planulae $^{13}$C CP, 2 weeks after the initial packing, showed a change in the disordered carbonate linewidth, suggesting that some mineral-phase maturation can occur in the animal post mortem. This indicates that the disordered surface phase is metastable and without live organism activity, crystallization may occur. The CP spectra of the planulae before and after 2 weeks and fresh planulae are compared in Supplementary Figure 10.

All ssNMR experiments were carried out at room temperature on a 11.74T Bruker Avance III spectrometer, and using a spinning rate of 10 kHz. All carbon-detecting experiments were preformed using a SPINAL64 heteronuclear decoupling at a field of 92 kHz during acquisition. Cross-polarization (CP) experiments were preformed using 2.5 μs of $^1$H 90° pulse and a contact time of 2 ms, 16,384 repetitions, and a recycle delay of 4 s. The CP block employed a ramped lock field on the protons between 100 kHz and 50 kHz and a carbon lock field of 65.8 kHz. The direct excitation (DE) measurements were preformed using 300 repetitions in signal acquisition and were carried out using 1.7 μs of $^{13}$C 40° pulse and a recycle delay of 240 s to accommodate the longitudinal relaxation time of the

crystalline $CaCO_3$ phases. 2D $^1H$–$^{13}C$ HETCOR experiments were preformed using 32 points in the $^1H$ homonuclear decoupled evolution time ($t_1$) and 2048 points during carbon evolution ($t_2$), 4000 repetitions, and a recycle delay of 1 s. The initial $^1H$ 90° excitation and CP step were performed with similar power levels, as mentioned in the CP experiment above. Homonuclear decoupling using the PMLG sequence (PMLG5 with supercycling) at an effective field of 105 kHz was used during $t_1$ with an acquisition time of 363 μs along this dimension. The 2D $^{13}C$ DARR experiments employed a similar $^1H$ 90° excitation pulse and field strengths in the CP preparation step, identical to the ones reported above for the CP experiment. The spectra were collected with 152 and 2048 $t_1$ and $t_2$ points, respectively, with 1536 repetitions and a recycle delay of 1 s. A heteronuclear decoupling field of 92 kHz was used during $t_1$ evolution. A $^1H$ irradiation field of 10 kHz was used in the 100-ms mixing time to fulfill the rotary resonance with the sample spinning at 10 kHz.

**Data availability**. The authors declare that the main data supporting the findings of this study are available within the article and its Supplementary Information files. Extra data are available from the corresponding author upon request.

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

## Acknowledgements

We thank Nico Sommerdijk for reading the manuscript and suggesting improvements, Lia Addadi and Steve Weiner for providing access to the cryo-SEM and micro-Raman facilities, and for the fruitful discussions. T.M. acknowledges support from the Israel Science Foundation (Grant 312/15), United States-Israel Binational Science Foundation (BSF; Grant # 2014035), and from the European Research Commission (ERC; Grant # 755876). A.A. is an awardee of the Weizmann Institute of Science––National Post-doctoral Award Program for Advancing Women in Science.

## Author contributions

A.A., I.P., G.G. and T.M. carried out the experiments, participated in the data analysis, participated in the design of the study, and drafted the manuscript; M.N. carried out the field work and qPCR analysis; I.P., A.A., M.N. and T.M. carried out the Raman analysis and helped draft the manuscript; A.A. and T.M. carried out the cryo-SEM imaging; A.A. and K.K. carried out the cryo-EDS analysis; and R.G. and G.G. carried out the ssNMR experiments and data analysis. All authors gave their final approval for publication.

## Additional information

**Competing interests:** The authors declare no competing interests.

