## [Peer Review File · Nature Communications]

Editorial Note: This manuscript has been previously reviewed at another journal that is not operating a transparent peer review scheme. This document only contains reviewer comments and rebuttal letters for versions considered at Nature Communications. Mentions of prior referee reports have been redacted.

Reviewers' Comments:

Reviewer #1:

Remarks to the Author:

I have now reviewed this article for the third time, and believe that it is significantly improved as compared to the previous submission. The authors have absolutely done their best to address my comments.

While I didn't think that the article quite achieved the very high standards demanded for Nature Materials, I do believe that it is of a quality suitable for publication in Nature Communications, and thus recommend publication subject to a few (small) points listed below.

There are some very nice findings in the study - particularly the change from Mg to Sr rich environments during the development of the polyp, and the analysis of the expressed proteins - which make it significant to those working on corals, and biomineralization in general.

P5. The authors suggest that Mg stabilizes ACC while Sr promotes aragonite. It is well known that Mg also supports aragonite formation. The authors therefore need to cite supporting publications if they wish to state that Sr is more effective than Mg in promoting aragonite.

P7. Line 185. "... from slightly disordered aragonite, or proto aragonite". These are two entirely different materials. Slightly disordered aragonite would be crystalline, while proto-aragonite is the term used for ACC which exhibits some short range structure resembling aragonite. One is crystalline and the other isn't. Can their data not be used to distinguish between these very different materials?

Reviewer #2:

Remarks to the Author:

The manuscript is concerned with the first stages of the biomineralization of scleractinian corals. As correctly said by the authors, this stage is usually described as a non mineralizing stage, and new data will be helpful for a better understanding of the calcifying process. Because of the small size of the sample, such a study is difficult; moreover, the "soft" or very weakly calcified material is not well suited for the techniques used to characterize the mineral.

Despite its potential interest, the paper suffers from some problems.

First, only one species was studied. It is easy to understand why, because of the technical problems. But it is not in accordance with the title, in which one finding is said to be right for all corals.

L32: Abstract/Introduction: What are the "mature aragonitic pillars characteristic of the stony coral skeletons"? I do not see "pillars" in the usual literature about the structure of corals.

L41-43: "Skeletal growth, in adult corals, involves transformation of amorphous calcium carbonate precipitate into an aragonite-made skeleton from centers of calcification that contain a skeletal organic matrix (SOM)^{1,2}."

The cited references (1, 2) used to justify this affirmative sentence come from the lab in which one of the authors of the present manuscript was working. Moreover, it was the same species (*Stylophora pistillata*). All these data have not yet been studied in other species, and they are not yet confirmed.

L69-70: "We applied a multi-disciplinary approach that utilizes state-of-the-art spectroscopic, imaging, and molecular biological techniques..."

I have never seen people saying they use "old techniques". The positive point is the multi technique approach.

L84...: the nano-granules are not well visible in the figures. Only an enlargement of Fig. 2H on the screen shows them. Fig. 2e: no scale. It is very probable that the granular aspect of the EDS map is not due to the granular structure, but to the techniques itself. So many details about the set up are missing!

EDS analyses: what about nitrogen? C and O exist in calcium carbonate minerals. Only N is indicative of the presence of proteins.

L144: it is true that the Mg content is higher in COC than in the fibres. It has been shown in about 15 species (Cuif & Dauphin 1998). Nevertheless, 15 species are not a huge number. Another point is to note that in this study, it is shown that the Mg content is not significantly different in COC and fibres. Again, the authors over interpret and overgeneralise their findings.

L155-165, Fig. 3: the morphology of the crystals shown in the right part of the Fig. 3G image is very unusual for biological crystals. Up to now, all the SEM and AFM have shown that the calcareous biogenic crystals are made of irregular non-angular granules. No facets, no angle. Geometric morphologies are often seen in fossil samples, indicative of a re-crystallization process. Here, an artefact? Cryo-SEM is not perfect, and some problems are known.

It must be added that the irregular rounded nanogranules have been described in both aragonitic and calcite biogenic crystals, and in ACC of some gastroliths.

L318-323: the presence of a mineral is a "novel finding", but the rest of the paragraph is only verbose. Useful for a highlight?

L337...: the authors are right: proteins are involved in the mineralization process. But they are not alone. Sugars and lipids also play a role. The authors have no data about these organics, but their possible role must be evoked in the conclusion.

L423...: while the experimental conditions of the cryo EDS are not sufficient, numerous details about the raman are not necessary. Only the experimental data are needed, not the notice of the system;

General comments:

Despite the interest of the finding, only one species is studied, so the title and general conclusion are overstated. This manuscript must be submitted to a more specialized journal, dealing with corals. The citations of the data of the literature are sometimes biased, and the descriptive terms are not those used in the literature of the structure of corals. Some structural details are not clearly visible in figures.

Reviewer #4:

Remarks to the Author:

The article provides evidence of mineralisation occurring in free-swimming coral planulae. A previously unreported phenomenon in corals. The authors have addressed most of my initial concerns.

A few comments / questions:

Text: Coral development involves an initial, non-mineralized planktonic (free-swimming) stage where it develops from a long and narrow larva (Fig. 1A) into a spheroid planula (Fig. 1B), and subsequently into a benthic (settled) primary polyp phase (Fig. 1C)...

Comment: corals do not 'develop' from elongated into spherical. In fact, their shape changes constantly as they swim about (see for example, Titlyanov et al 1998, Marine Biology; Rinkevich & Loya 1979, Article I: Marine Ecology Progress Series). Shape cannot be used as an indicator of coral planulae development.

Text: For the day old pre-settled metamorphosed planulae, cryo-SEM shows extracellular deposits with sizes varying from one to tens of microns, identified as a mineral by backscattered electron (BSE) signals...

Comment: strictly speaking the BSE image does not allow you to determine the structure is a mineral/mineralised. Rather the image simply allows you to see a region that has a higher average atomic composition than the surrounds.

Fig 1 B: Why are the two planula so different in appearance? This is not normal. The one on the right has very few zooxanthellae and looks to be dying / dead.

Fig 2B: According to the image labels on Figure 2B, the crystalline deposit is located within the ectodermal cell layer. Yet the authors state that they were '... randomly found within the organic material of the endoderm and lipidic region...'. This needs to be clarified.

Detailed, point-by-point response to reviewers' comments

In the text below, the reviewers' comments are in black and our responses are in blue text. Citations from the revised manuscript appear in *italics blue* within quotation marks and yellow highlight represents text changed between original and revised manuscript.

Reviewers' comments:

Reviewer #1 (Remarks to the Author):

I have now reviewed this article for the third time, and believe that it is significantly improved as compared to the previous submission. The authors have absolutely done their best to address my comments.

While I didn't think that the article quite achieved the very high standards demanded for Nature Materials, I do believe that it is of a quality suitable for publication in Nature Communications, and thus recommend publication subject to a few (small) points listed below.

There are some very nice findings in the study - particularly the change from Mg to Sr rich environments during the development of the polyp, and the analysis of the expressed proteins - which make it significant to those working on corals, and biomineralization in general.

P5. The authors suggest that Mg stabilizes ACC while Sr promotes aragonite. It is well known that Mg also supports aragonite formation. The authors therefore need to cite supporting publications if they wish to state that Sr is more effective than Mg in promoting aragonite.

We suggest that Mg ions together with glu-rich CARP proteins stabilize ACC and that Sr ions with asp-rich CARP proteins promote aragonite formation. There are no works in the literature that indicate such a combined effect. The ions on their own, show a tendency towards mineralization in one form or the other. The literature on Mg effect is much wider than Sr and we certainly do not have opposing claims to published works. However, it is known from numerous papers, that Mg can lead to ACC, aragonite or Mg-calcite depending on Mg concentrations and other conditions. It is also usually noted that aragonite crystals do not contain Mg ions as impurities. For that matter, it is highly unlikely that Sr will be found in aragonite as well. Nevertheless, high Sr concentrations were reported in early CaCO₃ precipitation in several marine organisms, and calcareous Sr serves as an indicator of seawater composition in which the skeletons were formed. We cited two exemplary papers reporting these intriguing insights in the manuscript [refs 26,27]. In the paper by Kitano et al, [ref 27] there is very preliminary evidence for the disparate role of the two ions however, the paper is focused on the mere incorporation of high levels of Sr (and Ba) into early CaCO₃ precipitates rather than the tendency to form one polymorph or another. To the best of our knowledge, there is no comparative work done yet on the role of Mg vs Sr ions on stabilizing amorphous phase vs mature crystalline phase, or on polymorph selection.

P7. Line 185. "... from slightly disordered aragonite, or proto aragonite". These are two entirely different materials. Slightly disordered aragonite would be crystalline, while proto-aragonite is the term used for ACC which exhibits some short range structure resembling aragonite. One is crystalline and the other isn't. Can their data not be used to distinguish between these very different materials?

We thank the reviewer for that comment, to avoid confusion we removed the term proto-aragonite throughout the text.

Reviewer #2 (Remarks to the Author):

The manuscript is concerned with the first stages of the biomineralization of scleractinian corals. As correctly said by the authors, this stage is usually described as a non mineralizing stage, and new data will be helpful for a better understanding of the calcifying process. Because of the small size of the sample, such a study is difficult; moreover, the "soft" or very weakly calcified material is not well suited for the techniques used to characterize the mineral.

We agree with the referee that it was challenging to obtain the data presented here, but primarily due to the low quantities of mineral and proteins present in the early stages. However, the NMR and Raman techniques are fully capable of detecting and characterizing soft and hard materials. It is one of the specialties of these techniques to determine order in condensed phases in the form of spectral line widths, spectral line position, and most of all the existence of lines that are characteristic of crystallized materials showing long range order. The combination of different microscopic, spectroscopic and biochemical characterization techniques is one of the unique features of this study, providing the first complete picture of coral larva calcification in *S. pistillata*.

Despite its potential interest, the paper suffers from some problems.

First, only one species was studied. It is easy to understand why, because of the technical problems. But it is not in accordance with the title, in which one finding is said to be right for all corals.

We thank the reviewer for this comment, we modified the title to be more specific to the studied species *S. pistillata*. Nevertheless, we would like to note that it is one of the most studied species in coral biology, with more than 7200 peer reviewed manuscripts. This highly characterized species is used by many top labs as a model organism, and thought to represent corals in general.

L32: Abstract/Introduction: What are the "mature aragonitic pillars characteristic of the stony coral skeletons"? I do not see "pillars" in the usual literature about the structure of corals.

We thank the reviewer for this comment, we used the word pillar to describe the orthorhombic/columnar shape that typical of aragonite micro structure of the coral skeleton. We modified the description throughout the text to the more common description in the literature: "*aragonite fibers*".

L41-43: "Skeletal growth, in adult corals, involves transformation of amorphous calcium carbonate precipitate into an aragonite-made skeleton from centers of calcification that contain a skeletal organic matrix (SOM)^{1,2}."

The cited references (1, 2) used to justify this affirmative sentence come from the lab in which one of the authors of the present manuscript was working. Moreover, it was the same species (*Stylophora pistillata*). All these data have not yet been studied in other species, and they are not yet confirmed.

We have modified the sentence to emphasize that the recent findings were related to *S. pistillata*. The sentence now reads: "*Recently it has been shown that skeletal growth in the adult coral *Stylophora pistillata*, involves transformation of amorphous calcium carbonate precipitate into an aragonite-made skeleton from centers of calcification that contain a skeletal organic matrix (SOM)^{1,2}.*"

L69-70: "We applied a multi-disciplinary approach that utilizes state-of-the-art spectroscopic, imaging, and molecular biological techniques..."

I have never seen people saying they use "old techniques". The positive point is the multi technique approach.

We have removed the "state of the art" statement.

L84...: the nano-granules are not well visible in the figures. Only an enlargement of Fig. 2H on the screen shows them. Fig. 2e: no scale. It is very probable that the granular aspect of the EDS map is not due to the granular structure, but to the techniques itself. So many details about the set up are missing!

EDS analyses: what about nitrogen? C and O exist in calcium carbonate minerals. Only N is indicative of the presence of proteins.

We thank the reviewer for this comment, we clarify the text indicating that the nano-granule structure can be seen in the Cryo-SEM image both in fig 2D and in higher resolution in the supplementary image of S1. In the previous review round, we were asked to move the higher resolution images to the Supplementary section. The Cryo-EDS only shows the elemental composition and not the nano-granular morphology. The granular form of the EDS image is mainly due to the lower resolution of the technique, made evident when trying to match images between the SEM and the EDS.

The sentence now reads "*These deposits are randomly found within the organic material of the endoderm and lipidic region (see Fig. 2A). High-resolution images (Figs. 2D and S1) reveal that these deposits have a nano-granular structure. Additionally, the cryogenic energy-dispersive X-ray spectroscopy (cryo-EDS) (Figs. 2E-2F) confirm the presence of elevated concentrations of calcium and carbon, and a depleted level of oxygen relative to the surrounding.*"

A scale bar was added to Fig. 2E and Fig. 3E.

We are not sure which details of the EDS setup are missing from the description as the reviewer commented. In the methods section we fully describe the setup, this includes the instrument description, the working distance, the energy used for the measurement and the software for the final analysis. We believe that this is all the information that is needed in order to repeat such an experiment, but would be happy to add additional information would the reviewer request so.

The Methods section reads: "*Elemental analysis and mapping were carried out at the same location by an EDS detector placed on the same optical axis as the SEM microscope. The loci of interest underwent cryo-EDS analysis using the microscope at a working distance of 6.5mm, the spot size of 300nm, dwell time of 8µs and an acceleration voltage of 8 kV, using a Bruker Quantax microanalysis system with an AXSFlash® detector. Element distribution maps were obtained using the Quantax software. The Identification of the Ca, Mg, Cl, C, O, Na is based on the K edge, while the Sr identification is based on the L edge.*"

Regarding a nitrogen peak in EDS: Nitrogen, although an important indicator of proteins is not as abundant a species compared to carbon and oxygen. Hence it may well be at or below the limit of detection. Additionally, the small N peak in EDS at 0.392keV falls right in between the more dominating carbon (-0.277keV) and oxygen lines (-0.525keV) and is likely masked by those two elements.

However, the information that the mineral phase contains proteins comes from the detailed NMR analysis, which is more sensitive to this fact than the EDS. At the same time, the presence of Mg and Sr is verified by the EDS, a method that is more suitable for such information. The combination of the two methods brings the comprehensive understanding of the detailed composition in each developmental stage of the coral.

L144: it is true that the Mg content is higher in COC than in the fibres. It has been shown in about 15 species (Cuif & Dauphin 1998). Nevertheless, 15 species are not a huge number. Another point is to note that in this study, it is shown that the Mg content is not significantly different in COC and fibres. Again, the authors overinterpret and overgeneralise their findings.

To avoid the impression that we are overgeneralizing our findings we have – in addition to stating that our results are for one species only – changed to text to read:

"Mg ions were shown to be distributed non-uniformly with particularly high concentration in the COC of more than 15 species^{1, 2, 29}. We do however note that Cuif & Dauphin 1998 found Mg in more than 15 coral species from diverse climates and locations, suggesting it is certainly not an uncommon phenomenon in corals.

L155-165, Fig. 3: the morphology of the crystals shown in the right part of the Fig. 3G image is very unusual for biological crystals. Up to now, all the SEM and AFM have shown that the calcareous biogenic crystals are made of irregular non-angular granules. No facets, no angle.

Geometric morphologies are often seen in fossil samples, indicative of a re-crystallization process. Here, an artefact? Cryo-SEM is not perfect, and some problems are known.

It must be added that the irregular rounded nanogranules have been described in both aragonitic and calcite biogenic crystals, and in ACC of some gastroliths.

Sharp crystals are not as uncommon as the reviewer claims. Examples of CaCO₃ crystals in living organisms that show sharp edges in cryo-SEM images after a freeze fracture preparation have been shown in e.g. in pteropods and foraminifera, (Figure 1 and 2 at Addadi and Weiner 2014 *Physica Scripta Phys. Scr.* 89 (2014)). As the samples are prepared by high pressure freezing, the recrystallization of the CaCO₃ under temperature conditions (< -120 °C), is not a realistic scenario.

L318-323: the presence of a mineral is a "novel finding", but the rest of the paragraph is only verbose. Useful for a highlight?

We have modified the paragraph to highlight our finding and conclusion but to avoid the redundancy of the paragraph. The paragraph is now reads: "*We show that during the coral *S. pistillata* development, mineral is formed while the planula is still in the pre-settled metamorphosed stage, which is earlier than what is currently thought. Having disordered mineral (ACC) distributed in the animal body, may explain the rapid calcification of coral skeleton at its most vulnerable phase, the settlement. Moreover, by using a combination of molecular biological tools, advanced electron microscopy, micro-Raman spectroscopy and NMR spectroscopy, we show the involvement of different proteins and ions in the two different developmental stages of the planula, exposing with detail the intimate interaction of the mineralization protein CARP2 with the ACC and the intimate binding of another set of active proteins (CARP1, CARP 3 and CARP4) to the aragonitic phase that develops upon settlement. We further note that the role of other biomolecules such as polysaccharides and lipids in calcification*

is yet to be determined. As in many other calcifying organisms^{11,43,44,45}, coral skeleton organic matrix contains also polysaccharides, and lipids^{46,47}. Evidence for the intimate interaction of other biomolecules with the mineral carbonates in the settled polyp is seen, for example, in the solid-state NMR cross peaks (Fig. 5D) between aragonite carbonates and aliphatic carbons at 92-96 ppm. However, the exact assignment of these carbons to lipids or polysaccharides is more complex and is outside the scope of the present study.

L337...: the authors are right: proteins are involved in the mineralization process. But they are not alone. Sugars and lipids also play a role. The authors have no data about these organics, but their possible role must be evoked in the conclusion.

We have evidence for involvement of other biomolecules, but restricted the discussion to biomolecules we could assign and compare the changes in, i.e. proteins. Nevertheless, we added in the discussion the finding about proximity to carbonates of lipids or polysaccharides:

"We further note that the role of other biomolecules such as polysaccharides and lipids in calcification is yet to be determined. As in many other calcifying organisms^{11,43,44,45}, coral skeleton organic matrix contains also polysaccharides, and lipids^{46,47}. Evidence for the intimate interaction of other biomolecules with the mineral carbonates in the settled polyp is seen, for example, in the solid-state NMR cross peaks (Fig. 5D) between aragonite carbonates and aliphatic carbons at 92-96 ppm. However, the exact assignment of these carbons to lipids or polysaccharides is more complex and is outside the scope of the present study."

L423...: while the experimental conditions of the cryo EDS are not sufficient, numerous details about the raman are not necessary. Only the experimental data are needed, not the notice of the system;

We have modified the methods section and have shortened the description of the Raman system. Please see the above response regarding the EDS description (L84).

General comments:

Despite the interest of the finding, only one species is studied, so the title and general conclusion are overstated. This manuscript must be submitted to a more specialized journal, dealing with corals. The citations of the data of the literature are sometimes biased, and the descriptive terms are not those used in the literature of the structure of corals. Some structural details are not clearly visible in figures.

We feel – with reviewer 1 and 4 – that our results are significant to those working on corals, and biomineralization in general. By taking into account all comments of the referee we trust this study is suitable for publication in Nature Communications.

Reviewer #4 (Remarks to the Author):

The article provides evidence of mineralisation occurring in free-swimming coral planulae. A previously unreported phenomenon in corals. The authors have addressed most of my initial concerns.

A few comments / questions:

Text: Coral development involves an initial, non-mineralized planktonic (free-swimming) stage where it develops from a long and narrow larva (Fig. 1A) into a spheroid planula (Fig. 1B), and subsequently into a benthic (settled) primary polyp phase (Fig. 1C).

Comment: corals do not 'develop' from elongated into spherical. In fact, their shape changes constantly as they swim about (see for example, Titlyanov et al 1998, Marine Biology; Rinkevich & Loya 1979, Article I: Marine Ecology Progress Series). Shape cannot be used as an indicator of coral planulae development.

We have changed the description of the planula development and added the above citation.

The paragraph now reads: *"Coral development involves, two non-mineralized planktonic (free-swimming) stages; one in the first day after spawning, the planula swims actively and frequently changes its shape from spherical to pear-like, disc-like and rod-like shape (Fig. 1A). The second stage is a metamorphosed globe shaped mature larva which has 6 pairs of complete mesenteries (Halcampoides stage) (Fig. 1B). This larva subsequently changes into a benthic (settled) stage of a primary polyp phase (Fig. 1C) in which the formation of the aragonite exoskeleton is thought to initiate⁶⁻⁸"*

Text: For the day old pre-settled metamorphosed planulae, cryo-SEM shows extracellular deposits with sizes varying from one to tens of microns, identified as a mineral by backscattered electron (BSE) signals...

Comment: strictly speaking the BSE image does not allow you to determine the structure is a mineral/mineralised. Rather the image simply allows you to see a region that has a higher average atomic composition than the surrounds.

We have modified the statement in order to clarify the BSE images. The sentence is now reads "For the day old pre-settled metamorphosed planulae, cryo-SEM shows extracellular deposits with sizes varying from one to tens of microns. When these structures are identified by more intense backscattered electron (BSE) signals, it indicates that these deposits consist of elements of higher electron density, possibly calcium"

Fig 1 B: Why are the two planula so different in appearance? This is not normal. The one on the right has very few zooxanthellae and looks to be dying / dead.

We thank the reviewer for this comment, we have changed the image of the metamorphosed planula.

Fig 2B: According to the image labels on Figure 2B, the crystalline deposit is located within the ectodermal cell layer. Yet the authors state that they were '... randomly found within the organic material of the endoderm and lipidic region...'. This needs to be clarified.

We thank the reviewer for this comment, we have clarified the description in the figure legend. The mineral particles are located in the endoderm and typically we see next to them lipid glands.

Reviewers' Comments:

Reviewer #2:

Remarks to the Author:

The manuscript is now more realistic, thanks to the new title (among other features).

The authors are right when they say that *Stylophora pistillata* is the most studied species. Nevertheless, variations exist in the large number of species, and it is not known how the colonial - single structure, the symbiotic - non symbiotic ... play a role in the early development.

Despite the rebuttal of the authors about the nanograins, they are not visible in fig. 2d and S1. To add a detailed view of fig. 1H in supplementary data will be helpful (even if it is an electron micrograph magnification). Moreover, some pictures are so dark that nothing is visible (2D, S1B and C, S2d for example). Similarly the banding patterns in fig. S1a and b can be removed with ordinary software. To improve the quality of these pictures is not to change the data.

I am not convinced by the explanation about the angular morphology of fig. 3G. High-pressure freezing and freeze substitution only minimize the artifacts. But I understand that the material is difficult and fragile, and the authors do their best.

I believe that the more interest of this paper is the presence of mineral at this early stage of growth. The role of the organic components is now accepted in the biomineralization field. As for other skeletons, the soluble organic matrices are an assemblage of hundreds of proteins, lipids and sugars, and there is also an insoluble complex matrix. It is not reasonable to try to explain the mineral secretion with 4 proteins.

Reviewer #4:

Remarks to the Author:

My concerns have been addressed.

Detailed, point-by-point response to reviewers' comments

In the text below, the reviewers' comments are in black and our responses are in blue text. Citations from the revised manuscript appear in *italics blue* within quotation marks and yellow highlight represents text changed between original and revised manuscript.

Reviewers' comments:

Reviewer #2 (Remarks to the Author):

The manuscript is now more realistic, thanks to the new title (among other features).

The authors are right when they say that *Stylophora pistillata* is the most studied species. Nevertheless, variations exist in the large number of species, and it is not known how the colonial - single structure, the symbiotic - non symbiotic ... play a role in the early development.

We are glad the reviewer finds the manuscript more realistic. Indeed it would be very interesting to understand how the process of skeletogenesis differs between corals, and this is certainly some of the future work we envisage emanating from this manuscript. However, such cross-species work is extremely labor intensive (especially the spectroscopic and microscopic analyses), likely to take several years at the very least, and is beyond the scope of this study.

Despite the rebuttal of the authors about the nanograins, they are not visible in fig. 2d and S1.

To clarify this point further, we added a digital enlargement in fig 2d in a box. The granular structure is clearly visible in figure S1 (C,F) mainly at the interface between the mineral and the organic matrix. Figure S1 C and F are enlargements of specific regions in images A and D respectively, as is indicated in the figure legend. In addition, we added arrows that point at the interface, where the granular nature of the mineral is clearly visible.

To add a detailed view of fig. 1H in supplementary data will be helpful (even if it is an electronic magnification).

Since there is no figure 1H, we assume that the reviewer means 2H. We clarify the legend of figure 2 in order to clearly mention that detailed view of image 2H can be seen in figure 2I. The region of 2I is indicated by an orange box in figure 2H.

“(G-I) A developing aragonite mineral embedded in the organic material of the tissue. (G) SEM image of a 10 μm aragonite crystal, it is digitally colored in green based on the back-scattered electron (BSE) image. (H) SE image of **the region depicted in an orange box** in G, showing the mineral. (I) Higher magnification of the region depicted in an orange box in H, showing that acicular aragonite crystals emerge from a nano-granular structure.”

Moreover, some pictures are so dark that nothing is visible (2D, S1B and C, S2d for example).

The images mentioned by the referee are all backscattered detector images (BSE). Such images give a dark signal unless high contrast objects are present. Similar to many other

studies (e.g., Khalifa, G. M. *et al. Journal of Structural Biology*, doi:<http://dx.doi.org/10.1016/j.jsb.2016.01.015> (2016), Prior et. al. *American Mineralogist* (1999) 84 (11-12): 1741-1759.), we use this detector to distinguish between different materials in the sample that have different Z values. Using this method, one can easily detect highly dense regions, which are in the case of mineralized skeleton – the mineral itself. Such regions then appear in light color on the dark background. Thus, BSE images should be dark otherwise one will not see the effect of the heavy ions on the image. Finally, we note image 2C is the complementary BSE image of figure 2B, and image 2D is the BSE image of the same region which is annotated by orange box in figure 2C. Thus, a “regular”, secondary electrons, image of this region is also presented. The same holds also for the images in the SI: figure 1B is the BSE complementary image of figure S1A, and S2D is the BSE complementary image of S2C where we prove that the brighter structures in S2C are indeed mineral.

We found that, for one image (S1C), improving the brightness and contrast resulted in a better image. We attached below the histogram of this image - any further modification will remove information from the image.

Similarly the banding patterns in fig. S1a and b can be remove with ordinary software. To improve the quality of these pictures is not to change the data.

The “banding pattern” is a charging affect that is caused due to the non-conductive nature of the sample (local charging). The sample is imaged in -120 Celsius without any metal coating. Following the reviewer suggestion we explored whether removing the charging effect using Matlab increased the clarity. The two image series are presented side-by-side below. We believe that the original figures, that have not been filtered, are much more informative, and that the filtered images have new artifacts that were not present prior to digital filtering. Nevertheless we leave it to the editor’s decision whether to use the filtered one over the original one. The figure legend below includes text that clarifies the filtering, if selected:

Figure S1: Mineral deposits at the pre-settled larval stage, detected by cryo-SEM. (A-F) Several different extracellular mineral deposits. (A, D) SE image of a mineral deposit. The mineral is ~10 μm or larger. (B, E) BSE image of the same region as in (A). (C, F) Higher magnification of the region delimited by an orange box in (A, D). The granular structure is visible at the interface between the mineral and the organic matrix (orange arrows). A custom mask Fourier filter was used to remove the charging effects in A and D.

The original image:

After Matlab processing:

I am not convinced by the explanation about the angular morphology of fig. 3G. High-pressure freezing and freeze substitution only minimize the artifacts. But I understand that the material is difficult and fragile, and the authors do their best.

We appreciate the reviewer understanding for the difficulty in handling these types of samples. We would like to emphasize that these samples **are not** freeze substituted samples! The samples are frozen and imaged under cryogenic conditions as it is written just below the image in the image caption. *Freeze-substitution* is never mentioned in the text!

I believe that the more interest of this paper is the presence of mineral at this early stage of growth. The role of the organic components is now accepted in the biomineralization field. As for other skeletons, the soluble organic matrices are an assemblage of hundreds of proteins, lipids and sugars, and there is also an insoluble complex matrices. It is not reasonable to try to explain the mineral secretion with 4 proteins.

As suggested by the reviewer, it is unlikely that the entire process of mineral deposition in corals can be explained by four proteins. Nevertheless, there is not a single mineralization process which is completely understood from start to end with each protein, lipid and polysaccharide's role deciphered. Yet, we continue to pursue an understanding of how the collectively called "organic matrix", be it the soluble one or the insoluble one, is able to achieve this complex mission with remarkable control. Here, we have brought the field a large step forward by associating four well-characterized proteins, which were discovered by analysis of mature coral skeleton (Mass et al. 2013, Current Biology, Drake et al 2013, PNAS), with the ligands they bind to and regulate in the context of the entire organism, and during early development. To the best of our knowledge, this is the first time correlation has been observed between different mineral phases and protein expression, as well as the first time that anyone has shown the involvement of specific proteins in biominerals at this level, especially as it is depicted in our NMR data. As we wrote in the text and in the response to reviewer #4 in the previous letter, we are not excluding the involvement of other organic agents in this process but it is beyond the scope of this paper and is a subject for many years of research. To further clarify this point, we have added a recent publication on polysaccharides as well (Naggi, A. *et al.* Structure and Function of Stony Coral Intraskelatal Polysaccharides. *ACS Omega* **3**, 2895-2901, doi:10.1021/acsomega.7b02053 (2018)).
From the discussion: "We further note that the role of other biomolecules such as polysaccharides and lipids in calcification is yet to be determined. As in many other calcifying organisms^{11,43,44,45}, coral skeleton organic matrix contains also polysaccharides, and lipids^{46,47,48}. However, the exact assignment of these carbons to lipids or polysaccharides is more complex and is outside the scope of the present study....."

Reviewers' Comments:

Reviewer #2:

Remarks to the Author:

nanograins are not yet visible for people convinced they do not exist (it is not my case). No new data are necessary, only an improvement of the contrast of the related pictures is asked for. It is not a difficult task using modern software.

It is my only request.